# An entropy-initiated coupled-trait ODE framework for modeling longitudinal cohort dynamics

Anderson M. Rodriguez[ID]*

The University of Georgia, Athens, Georgia, United States of America

* amr28693@uga.edu

## Abstract

This work introduces a minimal, information-theoretic dynamical framework for modeling longitudinal cohort data using an entropy-initiated system of coupled-trait ordinary differential equations (ECTO). For each survey wave, item-level Likert responses are compressed into a normalized Shannon entropy index that summarizes cross-sectional dispersion; this index is used to initialize the low-dimensional state variables of the autonomous ODE system. ECTO then tracks the interactions among a primary trait-like state, a secondary coupled state, and a latent environmental-stress component through phenomenological terms representing generic self-limitation, trade-offs, and feedback. Using data from the Swedish Adoption/Twin Study on Aging (SATSA), the framework reproduces broad cohort-level trajectories and is evaluated with leave-one-wave-out forecasting and comparisons against simple statistical baselines. A second longitudinal dataset of U.S. dental student data provides an external validation test, demonstrating that low-dimensional dynamics initialized from entropy measures can generalize across cohorts with different measurement instruments, demographic compositions, and timescales. Across both datasets, ECTO achieves stable out-of-sample performance, indicating that major cohort-level trends can be captured without assuming complex latent-variable models or time-varying causal inputs. Entropy here functions as a compact summary of population heterogeneity rather than a dynamical driver, and the coupled ODEs supply an interpretable alternative to high-dimensional or black box machine-learning approaches. This framework establishes a concise, transparent method for linking information-theoretic preprocessing with cohort-level dynamical modeling and provides a foundation for future multivariate or multi-cohort extensions.

## 1 Introduction

Longitudinal cohort studies generate rich, temporally spaced behavioral measurements, but the resulting data are often high-dimensional, categorical, and sparsely sampled. These properties make it difficult to directly apply continuous-time modeling

**Data availability statement:** All original raw data files are available from the original respective repositories: https://www.icpsr.umich.edu/web/NACDA/studies/3843 and https://www.esapubs.org/archive/ecol/E087/047/default.htm. Entropy-based datasets and a full results/reproduction-focused 'walkthrough' are available at the author's repository: https://github.com/amr28693/ECTO_walkthrough_2026.

**Funding:** The author(s) received no specific funding for this work.

**Competing interests:** The author has declared that no competing interests exist.

techniques. A central challenge is identifying low-dimensional structure that captures broad cohort-level trends while remaining interpretable and computationally tractable [1]. This work addresses that challenge by introducing a minimal coupled ordinary differential equation (ODE) framework, the Entropy-Initiated Coupled-Trait ODE (ECTO) system, for modeling cohort-level dynamics using information-theoretic preprocessing.

For each survey wave, item-level Likert responses are compressed into a normalized Shannon entropy [2] index that summarizes cross-sectional dispersion within the cohort. This index provides a compact, data-derived initialization for the ODE state variables; the system then evolves autonomously. Entropy is not treated as a mechanistic driver but as a stable summary of population heterogeneity that sets the initial conditions from which low-dimensional dynamics unfold. This allows categorical psychometric data to be translated into a form suitable for continuous-time modeling without imposing assumptions about individual-level processes.

To describe how cohort-level states change over time, we construct a set of coupled nonlinear ODEs whose terms represent generic interaction, saturation, and feedback motifs commonly used in minimal dynamical models. Saturating terms impose soft capacity limits, interaction terms permit moderated coupling between variables, and a latent environmental-stress component introduces a slowly varying context. These functional forms were chosen for mathematical parsimony, stability, and interpretability. The resulting system is low-parameter, phenomenological, and designed to capture broad cohort-level trajectories rather than individual-level variation.

We evaluate ECTO using data from the Swedish Adoption/Twin Study on Aging (SATSA) [3], a multi-decade longitudinal study. Model performance is assessed using leave-one-wave-out forecasting and comparisons against simple statistical baselines. To examine generalizability, we apply the same framework to an independent longitudinal dataset of U.S. dental students [4] collected with different instruments, demographic profiles, and timescales. Across both datasets, ECTO recovers large-scale cohort trends and produces stable out-of-sample predictions, demonstrating that entropy-initialized low-dimensional dynamics can serve as a flexible approximation for longitudinal behavioral data. In what follows, we distinguish between qualitative demonstrations of system behavior under illustrative parameterizations and quantitative evaluation using parameters estimated via constrained optimization.

This paper presents ECTO as a methods-focused contribution: an interpretable, information-theoretic coupled ODE framework for modeling cohort-level dynamics from behavioral data. Psychometric data serve here as a structured empirical substrate, but the approach is not limited to psychological applications. The combination of entropy-based initialization and transparent dynamical structure provides a general template for linking high-dimensional observational data to low-dimensional continuous-time models.

## 2  Methods for data cleaning and comparative approach analysis

Data used in this study were gathered from the Inter-University Consortium for Political and Social Research (ICPSR) [3]. The data originates from a publicly available

program of research in gerontological genetics known as the Swedish Adoption/Twin Study of Aging (SATSA, or, ICPSR 3483), which began in 1984. SATSA is a longitudinal study exploring genetic and environmental influences on aging, especially cognitive and personality traits. It does so by comparing twins reared apart (TRA) with those reared together (TRT). Data were gathered from mail-in questionnaires as well as through in person cognitive and physical testing. Only results from the questionnaires are under consideration for the models presented. Respondents were surveyed on questions including health status, personal histories, and questions regarding personality, habits, and attitudes. The SATSA data used in this study was accessed on June 11th, 2025 for research purposes. The data is anonymized, and no personally identifiable information was accessible during or after data collection.

For this presentation, data up until 2007 were analyzed. Participants at trial start were n ~ 1,700, while that number reduced to roughly one-third of that initial cohort by the end of the survey period. Considering that the survey took place over 20 years with no additional members being added to the survey, the decrease in responding population is expected as natural attrition due to age.

Survey samples were collected in six distinct waves (1984, 1987, 1993, 2003, 2007). *Appendix B* in S1 Appenidx demonstrates that entropy trends are significant and not artifacts of participant attrition over time. Survey questions were sampled by the author in service of refining the general ECTO model. A second dataset generated on a dental student cohort, from Leite et al [4] is used for external validation.

To explore the SATSA data in ECTO context, twelve psychometric questions were analyzed. These items were selected because they are consistently measured across waves and yield well-populated Likert distributions suitable for entropy calculations, e.g.,:

**P4 Worry:** "I often worry."

**P8 Hot-Temperedness:** "People think I am hot-tempered and temperamental."

**P9 Satisfaction:** "I often feel satisfied."

Note: The survey-internal designations, with some letter followed by a number, begin in this format starting in the 1984 frequency data from the SATSA [3] dataset. Subsequent years see the addition of non-alphabetically ordinal prefix letters (e.g., "V" is a prefix in 1987, and "K" is a prefix in 2007). (This note is included for purposes of future reproducibility regarding the conclusions of this paper.)

Each of the above traits was measured via a structured psychometric instrument [5,6] (the original Likert-scale survey in the SATSA study) with five sub-items per trait which explored the degree to which the subject felt they agreed with the presented question tied to an associated emotional state, characteristic, or phenomena ('exactly right'; 'almost right'; 'neither'; 'not quite right'; 'not right at all').

## 2.1 Utilizing Shannon entropy

Although raw Likert distributions were examined directly (see supplemental material *Appendix A* in S1 Appendix), the primary preprocessing step for the dynamical framework involved compressing each item's categorical response distribution into a single Shannon entropy value for each wave. For a given item and wave, the proportion of responses in each Likert category forms a discrete probability distribution, from which entropy $H(t)$ was computed in bits and later normalized for use in model initialization (Fig 1). Entropy here functions as a concise summary of cross-sectional dispersion within the cohort, capturing how broadly or narrowly responses are distributed at each measurement occasion.

Shannon entropy provides a principled measure of distributional uncertainty for categorical data and offers a consistent, interpretable way to summarize variation across waves [2,7]. Because each SATSA wave contains the same set of Likert-format items, entropy furnishes a comparable, scale-free measure of cohort heterogeneity across

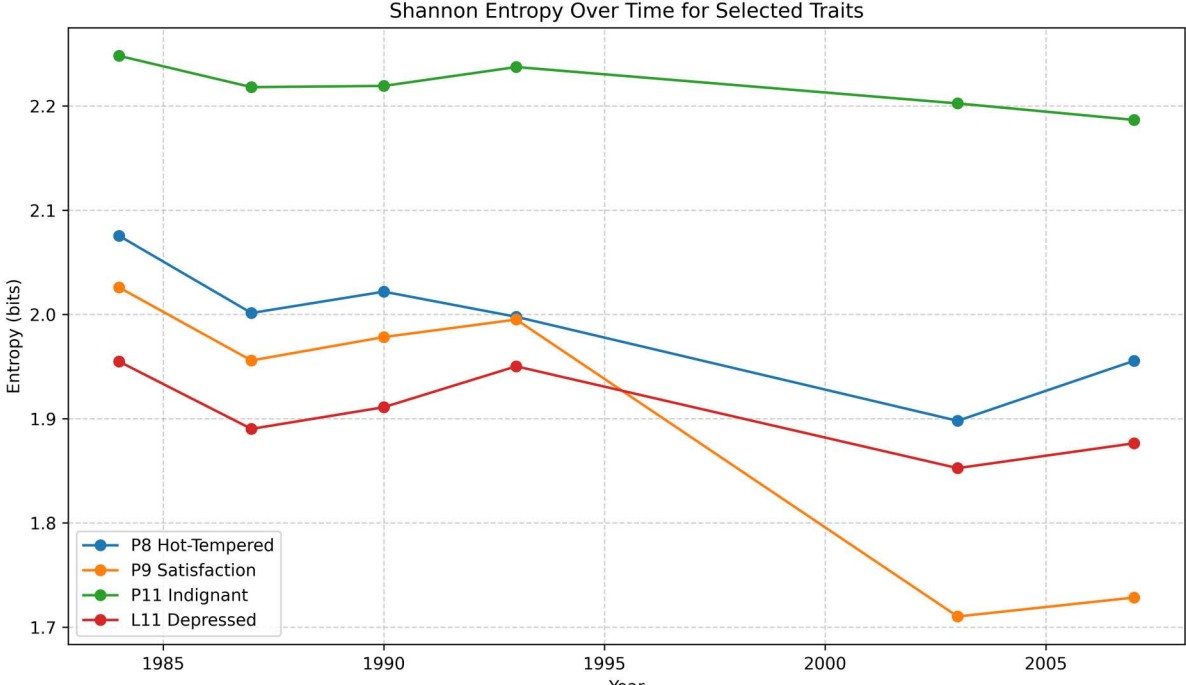

**Fig 1. Shannon entropy of four traits isolated from the set under review: Hot-Tempered, Satisfaction, Indignant and Depressed, across six SATSA survey years (1984–2007).**

time. In this study, entropy is used solely as a preprocessing step to generate low-dimensional initial conditions for the ODE system. The dynamical model itself evolves autonomously and does not treat entropy as a mechanistic or causal variable.

This preprocessing step was applied identically (see Repository: Track "B") to the Leite et al [4] dental student data-set used for external validation. In both datasets, entropy serves only as a compact representation of cross-sectional response patterns, enabling translation of high-dimensional categorical data into initial states for continuous-time modeling without imposing assumptions about individual-level processes or latent mechanisms.

## 2.2 Calculating and normalizing Shannon entropy

For each questionnaire item and survey wave, the categorical response distribution was summarized using Shannon entropy. Let $\vec{x}_t = [x_1, x_2, \ldots, x_n]$ denote the raw Likert category counts for a given item in wave $t$, where $n = 5$ response categories.

These counts are converted into probabilities

$$p_i = \frac{x_i}{\sum_{j=1}^{n} x_j},$$

yielding a discrete distribution $\vec{p}_t = [p_1, \ldots, p_n]$. The Shannon entropy for that item and wave is then

$$H_j(t) = -\sum_{i=1}^{n} p_i \log_2 p_i,$$

with the convention that terms with $p_i = 0$ are omitted. Entropy is measured in bits and reflects dispersion of responses across categories.

## 2.3 Normalization and construction of the pooled entropy index

Because different items may have different numbers of valid response categories across datasets, each item entropy is normalized by its theoretical maximum:

$$H_{j,\text{norm}}(t) = \frac{H_j(t)}{\log_2(n)} \in [0, 1].$$

To obtain a single cohort-level summary for each wave, normalized item entropies are averaged across the $J$ selected items:

$$H^*(t) = \frac{1}{J} \sum_{j=1}^{J} H_{j,\text{norm}}(t).$$

The resulting $H^*(t)$ is a dimensionless quantity on the unit interval $[0, 1]$ and represents the overall cross-sectional heterogeneity of the cohort at wave $t$. All entropy computations are performed independently for each wave and for each dataset (see *Appendix "A"* in S1 Appendix in the supplemental material for a worked example of calculating Shannon entropy from SATSA data).

## 2.4 Interpretation and use in the dynamical framework

The pooled entropy index $H^*(t)$ is a population-level summary statistic of response dispersion. It does not measure individual-level variability and is not interpreted as a psychological or biological state. In this modeling framework, $H^*(t)$ serves solely to initialize the low-dimensional ODE state at the first observed wave. After initialization, the dynamical system evolves autonomously without time-varying entropy inputs (in Appendix X we initialize a case with sinusoidal forcing for analytical contrast).

Time is measured in years, and the entropy measure $H^*(t)$ is dimensionless. This preprocessing procedure allows high-dimensional Likert data to be translated into stable, comparable initial conditions for continuous-time modeling while remaining agnostic to individual-level mechanisms.

## 2.5 Comparable approaches

Longitudinal psychometric data are typically analyzed using statistical frameworks such as structural equation modeling (SEM), mixed-effects models, and hierarchical Bayesian approaches [8–10]. These methods estimate latent constructs or individual-level trajectories but operate in discrete time and do not produce continuous-time system dynamics. As a result, they describe variation in observed traits but do not directly model how cohort-level states evolve under a generative dynamical law.

Within psychology, dynamical systems concepts have been invoked to describe developmental change or emotional regulation [11,12], yet these applications generally rely on qualitative state diagrams or symbolic representations rather than formal differential equations. Existing approaches therefore capture local or conceptual aspects of change but do not provide a continuous-time, low-dimensional model of cohort-level trait evolution.

Entropy-based techniques have also been used in behavioral science, e.g., in state-space grids [13,14], Markov transition models [15], and symbolic dynamics [16,17]. These methods quantify structure in time-ordered categorical sequences

but remain discrete and focus on fine-grained interaction patterns rather than population-level summaries across survey waves. None of these approaches integrate entropy into a continuous-time generative framework.

In contrast, ordinary differential equation (ODE) models are widely used in mathematical biology and other quantitative sciences to describe population-level dynamics in a transparent and interpretable manner [18,19]

While the above examples are at points conceptually similar to the ECTO method, to the author's knowledge, no existing framework converts longitudinal Likert-scale population data into normalized entropy indices and uses those indices to initialize a coupled, autonomous ODE system for cohort-level trait trajectories. The present approach is therefore methodologically adjacent to prior work in psychometrics, dynamical systems theory, and entropy-based behavioral analysis, but distinct in its use of pooled entropy as a compact initialization mechanism for continuous-time modeling.

## 3 Methods for the proposed ordinary differential equation system (ECTO)

This section describes the ECTO system, a set of low-dimensional coupled ordinary differential equations designed to generate smooth cohort-level trajectories from entropy-initialized starting values. The goal is not to model psychological or biological mechanisms, but to provide a transparent and mathematically tractable continuous-time approximation to population-level changes observed across survey waves.

ECTO treats the modeled quantities as abstract state variables whose dynamics are governed by phenomenological functional forms. These forms are chosen for their ability to represent saturating growth, cross-coupling, and bounded feedback within a low-dimensional system. The system is autonomous, and once initialized at the first wave using the pooled entropy index $H^*(t_0)$, it evolves solely according to its internal structure.

For clarity, the state variables and parameters are defined in Section 3.4. The full system is presented below.

### 3.1 State variable $N(t)$: Selection pressure dynamics

The primary state variable $N(t)$ evolves according to a selection–constraint balance:

$$\frac{dN}{dt} = \mu N - s(N) \cdot N,$$

(1)

where the selection pressure term is defined as

$$s(N) = \alpha N + \beta^2 P.$$

(2)

The linear term $\mu N$ represents a baseline influx or persistence of state magnitude at the cohort level. The selection pressure $s(N)$ is a phenomenological constraint capturing both self-limitation ($\alpha N$) and coupling to the companion state variable $P(t)$. The square on $\beta$ ensures that this coupling contributes a nonnegative damping effect regardless of parameter sign. These terms are not intended as mechanistic biological forces but as low-dimensional analogues of cohort-level constraint and interaction.

### 3.2 State variable $P(t)$: Pleiotropic dynamics

The companion state variable $P(t)$ evolves according to

$$\frac{dP}{dt} = \mu P - \beta P \cdot \left( \frac{E_{\text{metabolic}}}{G} \right),$$

(3)

where the metabolic cost term is defined as

$$E_{\text{metabolic}} = c_1 P + c_2 N + c_3 E_{\text{stress}}. \tag{4}$$

Here, $E_{\text{metabolic}}$ is an accounting variable that aggregates cohort-level costs associated with state magnitude, cross-state interaction, and accumulated environmental stress. The multiplicative structure ensures that costs scale with state magnitude, producing damped or saturating dynamics depending on parameter values. The normalization constant $G$ sets the effective capacity scale and is fixed for comparability across runs.

### 3.3 State variable $E_{\text{stress}}(t)$: Environmental stress dynamics

The state variable $E_{\text{stress}}(t)$ evolves according to

$$\frac{dE_{\text{stress}}}{dt} = \gamma E_{\text{stress}} \cdot \left( \frac{N}{N+K} \right). \tag{5}$$

This formulation allows stress to accumulate proportionally to existing stress levels, modulated by a saturating sensitivity to the primary state variable $N(t)$. The kernel $N/(N+K)$ prevents unbounded amplification at low state values and introduces a natural saturation scale. The system is fully autonomous after initialization.

### 3.4 Parameter definitions and model structure

- $N(t)$: primary entropy-initialized state variable
- $P(t)$: secondary coupled state variable
- $E_{\text{stress}}(t)$: auxiliary feedback state variable
- $\mu$: baseline influx or persistence rate shared by $N$ and $P$
- $\alpha$: self-limiting constraint coefficient for $N$
- $\beta$: cross-state coupling coefficient (squared in the $N$ equation to ensure nonnegative damping)
- $c_1, c_2, c_3$: weights defining the metabolic cost term $E_{\text{metabolic}}$
- $G$: normalization constant setting the effective capacity scale
- $\gamma$: amplification rate for the stress state variable
- $K$: positive saturation constant controlling stress sensitivity

All parameters are freely estimated during model fitting. They do not correspond to biological or psychological mechanisms; instead, they shape the qualitative behavior of the autonomous dynamical system. The structure is selected for interpretability, low dimensionality, and flexibility across datasets.

### 3.5 Initialization and simulation procedure

All simulations are initialized by setting $N(t_0) = H^*(t_0)$, using the pooled entropy value from the first observed wave. The remaining state variables $P(t_0)$ and $E_{\text{stress}}(t_0)$ are initialized using small positive constants or fitted baseline values. After initialization, the system evolves autonomously with no additional inputs.

This setup produces forward simulations: the ODE system is integrated through continuous time, and model outputs are compared to empirical observations at wave times. This approach evaluates whether a low-dimensional autonomous system, initialized using pooled entropy, can approximate observed cohort-level trends without requiring latent variables, discrete-time transitions, or explicit exogenous forcing.

## 3.6 Dimensionality and overfitting considerations

Because each trait is represented by a single pooled-entropy trajectory, the effective dimensionality of the input data is low. The ECTO system operates on these compressed signals rather than on the full set of item-level responses. Although the model contains several parameters, it functions as a continuous-time dynamical system rather than a pointwise regression, and its structure imposes inherent regularity through smoothness, coupling, and functional form.

Forward simulation from empirically initialized starting values provides additional constraint: once the initial condition is set, the system generates an entire trajectory without independently fitting each observation. As a result, the number of free parameters does not scale with the number of timepoints, and the system's behavior is shaped primarily by its global structure rather than by local adjustments at individual waves. This helps reduce the risk of overfitting while allowing the model to capture broad cohort-level trends.

## 4 Results

Before presenting the optimized model fits used for quantitative evaluation, we first illustrate the qualitative behavior of the ECTO system using a small number of hand-tuned parameter sets from the SATSA dataset [3]. These parameterizations are not intended for model selection or performance comparison, but to demonstrate that the proposed ODE structure produces stable, interpretable trajectories (see: Table 1) and responds to coupling and feedback terms in the expected manner (full parameter sweep results can be found in *Appendix C.2* of the supplemental materials in S1 Appendix). Quantitative fitting, validation, and model comparison are addressed separately using parameters estimated via constrained optimization. The following, Figs 2 and 3, are from a group of parameter sets depicted in the *Appendix D* of the supplemental material in S1 Appendix.

**Third parameter set (See Repository *Module A$_3$*):**

$$\alpha = 0.098, \quad \mu = 0.00001, \quad \beta = 0.17117, \quad \gamma = 0.03, \quad c_1 = 2.0, \quad c_2 = 0.21, \quad K = 0.5$$

For the fourth parameter set, the auxiliary variable $E_{\text{stress}}(t)$ was augmented with an explicit sinusoidal forcing term of the form $A\sin(\omega t)$, and the coefficient $c_3$ was used to scale the influence of this oscillatory component within the aggregated constraint term. Details of the forcing term are provided in the supplementary material (*Appendix C.2* in S1 Appenidx).

**Fourth parameter set (See Repository *Module A$_4$*):** Same as the second set in all parameters, but with an added sinusoidal forcing term $c_3$ term in the environmental stress component, set: $c_3 = 12.6$.

The present model produces (among other sets) the following results from a 'qualitative demonstration fit' Set 3:

**Table 1. Model Fit Evaluation Metrics Across Parameter Sets (See: *Repository: A$_1$ through A$_4$*.**

| Metric | Set 1 | Set 2 | Set 3 | Set 4 |
|---|---|---|---|---|
| *Hot-Tempered (N)* | | | | |
| RMSE | 0.4064 | 0.2117 | 0.1552 | 0.1590 |
| $R^2$ | -0.7206 | 0.5330 | 0.7491 | 0.7367 |
| Pearson *r* | 0.885 | 0.886 | 0.879 | 0.880 |
| DTW Distance | 1.854 | 0.834 | 0.648 | 0.655 |
| *Worry (P)* | | | | |
| RMSE | 0.2198 | 0.2205 | 0.2205 | 0.2915 |
| $R^2$ | 0.6118 | 0.6093 | 0.6095 | 0.3176 |
| Pearson *r* | 0.805 | 0.803 | 0.803 | 0.777 |
| DTW Distance | 0.712 | 0.722 | 0.725 | 0.783 |

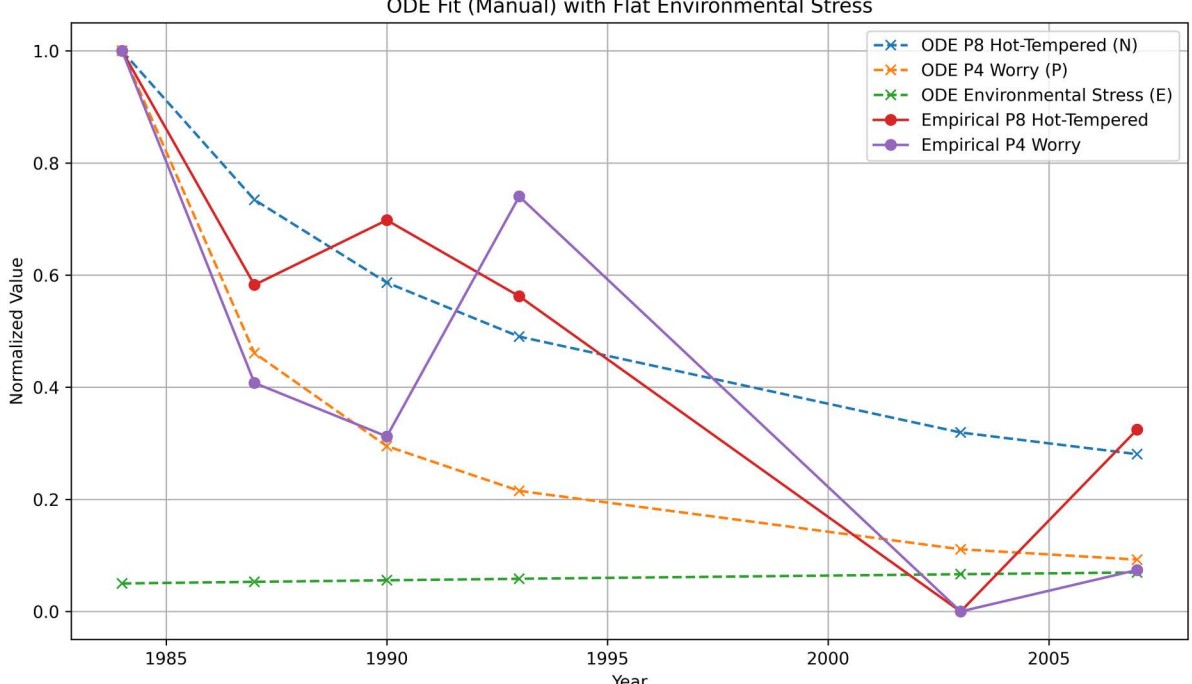

**Fig 2. ODE simulation with $\beta$ = 0.17117.** Fit to Hot-Tempered entropy improves further with higher $R^2$, while Worry entropy shows stable alignment with empirical data. This parameterization keeps the environmental stress term at baseline.

**Set 3 (Demonstrated Fit, see: *Repository:$A_3$*)**

$$\alpha = 0.098, \quad \mu = 0.00001, \quad \beta = 0.17117, \quad \gamma = 0.03,$$
$$c_1 = 2.0, \quad c_2 = 0.21, \quad c_3 = 0.0, \quad K = 0.5$$

State Variable (N) :   RMSE = 0.1552,   $R^2$ = 0.7491,   $r$ = 0.879 ($p$ = 0.021)

State Variable (P) :   RMSE = 0.2205,   $R^2$ = 0.6095,   $r$ = 0.803 ($p$ = 0.055)

**Set 4 (Oscillatory *E* Term, see: *Repository:$A_3$*))**

$$\alpha = 0.098, \quad \mu = 0.00001, \quad \beta = 0.17117, \quad \gamma = 0.03,$$
$$c_1 = 2.0, \quad c_2 = 0.21, \quad c_3 = 12.6, \quad K = 0.5$$

State Variable (N) :   RMSE = 0.1590,   $R^2$ = 0.7367,   $r$ = 0.880 ($p$ = 0.021)

State Variable (P) :   RMSE = 0.2915,   $R^2$ = 0.3176,   $r$ = 0.777 ($p$ = 0.069)

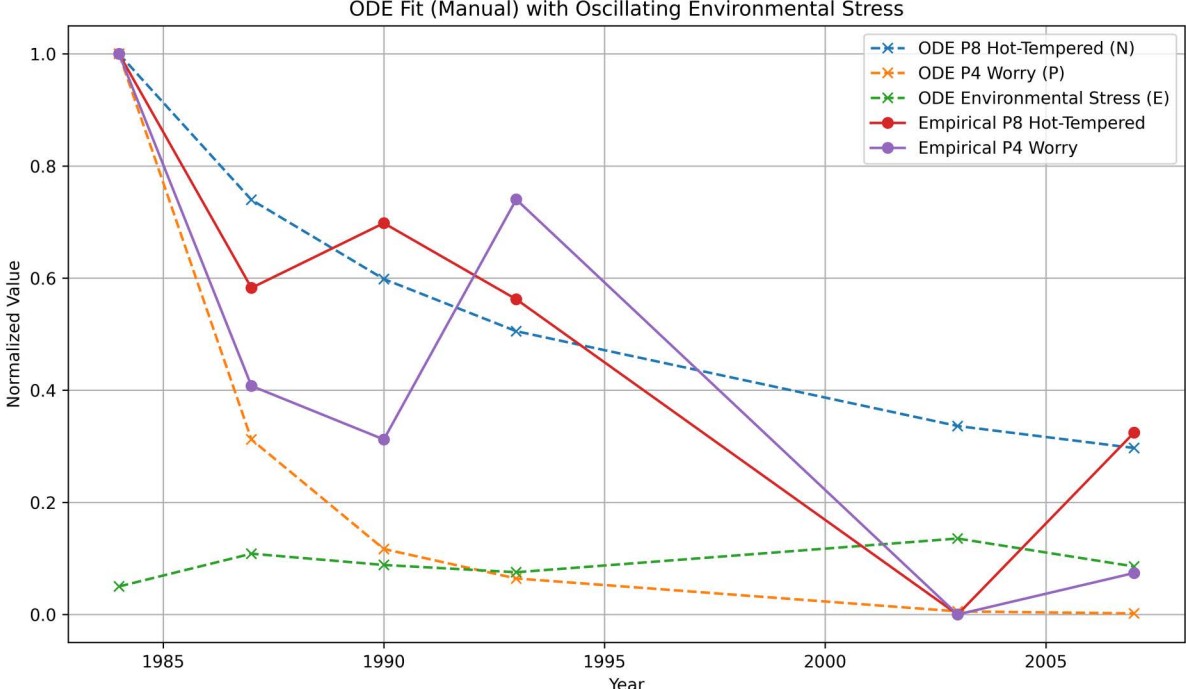

**Fig 3. ODE model with added oscillatory $c_3$ environmental stress term ($c_3$ = 12.6).** The fit shows notable influence from the Environmental stress term. Notably, 'Worry' entropy improves in demonstrative fit.

### 4.1 Entropy trajectories in the SATSA cohort

For the SATSA data, twelve psychometric items were converted into longitudinal entropy trajectories using the pooled entropy index $H^*(t)$. Across all traits, entropy values remained within the theoretical range for a five-category Likert scale (approximately 1.7–2.3 bits), and exhibited smooth, low-noise changes over the six survey waves.

Most traits (e.g., Fulfillment, Indignant, Curiosity, Life Optimism, Impulsivity, Excitement Preference, Rushed Feeling) showed modest gradual declines in entropy over time (see: *Figure* 2 in *Appendix A* of the supplemental material in S1 Appenidx), typically on the order of 0.05–0.15 bits from 1984 to 2007. A few traits (such as Competitive Ambition and Depressed) showed small non-monotonic fluctuations but remained within a narrow entropy band. One item, Satisfaction, displayed a slightly more pronounced drop between 1993 and 2003, followed by a mild rebound in 2007. Overall, the trajectories are continuous and well-behaved, providing suitable targets for continuous-time modeling (see *Repository: Module $A_0$*).

The flagship pair used in the ODE demonstrations, *Hot-Tempered* and *Worry*, also exhibited modest, interpretable variation. Hot-Tempered entropy declined slightly over the 23-year span with small deviations, while Worry entropy remained relatively stable around $\sim 2.15$ bits with only minor fluctuations. These patterns are sufficiently structured to test whether the ECTO system can generate smooth forward trajectories from entropy-initialized states without overfitting individual timepoints.

### 4.2 Forward simulation and fit for SATSA

For the SATSA flagship pair, the ECTO system was initialized using the first-wave entropy index and then integrated forward in continuous time. Parameters were estimated by minimizing the sum of squared errors between the simulated

trajectories ($N(t), P(t)$) and the normalized empirical entropy values for Hot-Tempered and Worry across all six waves, using a bounded L-BFGS-B optimizer.

A representative fitted parameter set yielded the following full-data metrics:

$$\text{RMSE}_N = 0.1541, \quad R^2_N = 0.7525$$

$$\text{RMSE}_P = 0.1896, \quad R^2_P = 0.7112.$$

These values indicate that, for both traits, the continuous-time model accounts for a substantial portion of the variance in the normalized entropy trajectories while maintaining relatively low residual error. Because all six timepoints are generated from a single integrated trajectory per variable, these metrics reflect the performance of the global dynamical structure rather than pointwise local adjustments.

## 4.3  Leave-one-wave-out validation for SATSA

To assess out-of-sample performance, a leave-one-wave-out (LOO) procedure was applied. For each of the six waves, the model was refit on the remaining five timepoints, and the held-out wave was predicted from forward simulation. The normalized empirical values and predictions for the Hot-Tempered and Worry trajectories showed moderate deviations but preserved overall trend shape.

Aggregated across holds, the LOO root mean square error was:

$$\text{LOO RMSE}_N = 0.1995, \quad \text{LOO RMSE}_P = 0.3056.$$

Given the small number of observations (six waves) and the autonomous, continuous-time formulation, these results indicate that the model retains reasonable predictive performance when individual waves are excluded from fitting. The errors are larger than in the full-data fit, as expected, but remain within a range consistent with the variability of the empirical entropy signals (see Repository *Module $B_6$*).

## 4.4  Comparison to an uncoupled baseline

As a structural baseline, an uncoupled variant of the system was considered in which the cross-term linking the two state variables was removed, and the variables evolved independently under simpler dynamics. This null system preserves the same initialization and general functional class but lacks coupling (see Repository *Module $A_{10}$*).

Across the SATSA flagship traits, the uncoupled baseline produced systematically higher RMSE values and lower $R^2$ than the coupled ECTO system. Although detailed numbers are reported in the supplementary material (see: *Appendix C.1* in S1 Appenidx), the overall pattern is consistent: allowing interaction between the state variables yields better alignment with the empirical entropy trajectories than modeling each trait index in isolation. This suggests that the coupling terms contribute identifiable structure beyond what can be captured with independent one-dimensional dynamics.

## 4.5  Cross-cohort application: Dental student dataset

To test the framework on a distinct cohort with different content, timescale, and sample size, the same modeling structure was applied to a longitudinal dataset of U.S. dental students [4]. Two psychometric items were selected: one reflecting perceived support (*supp*) and one reflecting perceived time pressure (*time*). For each item and wave (D1–D4), response distributions were converted to entropy values.

The resulting entropy trajectories were:

$$\text{supp}: 1.7278, \ 1.7198, \ 1.7755, \ 1.8126$$

$$\text{time}: 1.4530, \ 1.6855, \ 1.8210, \ 1.7319.$$

As with SATSA, the first-wave entropy values were used to initialize $N(t)$ and $P(t)$ (mapped here to *supp* and *time*, respectively), and the same ECTO structure was employed. Parameters were estimated under constrained optimization (see Repository: *Modules $B_3$, $B_4$, and $B_5$*).

A fully optimized fit (with all free parameters active) produced:

$$\text{RMSE}_{\text{supp}} = 0.2249, \quad R^2_{\text{supp}} = 0.6918$$

$$\text{RMSE}_{\text{time}} = 0.2406, \quad R^2_{\text{time}} = 0.5759.$$

A reduced-parameter fit, holding a subset of coefficients fixed and optimizing a smaller set ($\mu, \alpha, \gamma, E_0, \beta, c_1$), yielded similar performance:

$$\text{RMSE}_N = 0.2361, \quad R^2_N = 0.6605$$

$$\text{RMSE}_P = 0.2359, \quad R^2_P = 0.5922.$$

These results indicate that the same dynamical form used for SATSA can be transferred to a different cohort with different items and wave spacing, achieving moderate-to-strong alignment with the observed entropy trajectories under reasonable parameter settings.

### 4.6 Leave-one-wave-out validation for the dental dataset

The Dental dataset was also subjected to leave-one-wave-out validation. With four waves, each LOO run fit the model on three timepoints and predicted the held-out wave via forward simulation. The resulting LOO errors were:

$$\text{LOO RMSE}_N = 0.2903, \quad \text{LOO RMSE}_P = 0.2800.$$

Wave-level inspection showed near-exact predictions for some waves and larger deviations for others (notably D2), but overall the model preserved the qualitative trajectory shapes for both items (see Repository: *Module $B_6$*). Considering the very small number of timepoints, these errors are consistent with the level of variability in the entropy signals and suggest that the model generalizes across waves without catastrophic degradation.

### 4.7 Parameter sensitivity and multistart analysis

To probe local identifiability and robustness, a series of sensitivity tests and multistart optimizations were conducted, particularly for the Dental dataset where the number of parameters and timepoints are comparable.

For the reduced-parameter Dental fit, each parameter was perturbed by $\pm10\%$ around the optimized value while refitting as needed, and changes in total RMSE were recorded. The results indicated that:

- Some coefficients (e.g., those controlling overall growth and feedback strength) produced substantial changes in RMSE (up to $\sim 30$–$90\%$ increases), indicating strong local sensitivity and meaningful constraint by the data.

- Other coefficients (such as one of the secondary weights) had negligible effect on RMSE under $\pm 10\%$ perturbations, suggesting weaker identifiability in this small-sample setting.

A multistart procedure, in which the optimizer was initialized from multiple random seeds, produced a tight cluster of solutions for the reduced fit, with:

$$\text{mean SSE} \approx 0.488, \quad \text{sd} \approx 0.0001, \quad \text{min} \approx 0.4881, \quad \text{max} \approx 0.4885.$$

The narrow range of SSE across seeds indicates that the optimization landscape is relatively well-behaved and that the fitted parameter set is not an isolated artifact of a particular initialization.

For the SATSA flagship pair, an analogous analysis (reported in the supplementary material, *Appendix C*.2 in S1 Appenidx) also showed stable fits under multistart runs and modest local sensitivity for most parameters, consistent with the degree of information available in six-wave entropy trajectories.

### 4.8 Overfitting considerations in light of the empirical results

Given the small number of timepoints (six for SATSA, four for Dental) and the use of a nonlinear dynamical system, over-fitting risk must be considered. Three features of the present results help mitigate this concern:

1. **Entropy Compression:** The model operates on pooled entropy indices rather than raw item-level data, dramatically reducing the effective dimensionality of the input and forcing the system to account for broad distributional trends rather than individual responses.

2. **Autonomous Forward Simulation:** Once initialized at the first wave, the system generates a continuous trajectory using a fixed parameter set. Observed data at later waves are not individually targeted; instead, the model is evaluated on how well a single integrated curve aligns with all timepoints simultaneously.

3. **Cross-Dataset and LOO Performance:** Comparable performance levels on both SATSA and Dental datasets, together with the LOO results for each, indicate that the model's behavior is not limited to a single cohort or dependent on including every observation in fitting.

While the number of parameters is nontrivial relative to the number of timepoints, the use of a constrained functional form, entropy compression, and out-of-sample checks makes the present results consistent with a low-dimensional, interpretable model rather than a highly flexible, over-parameterized model. A more detailed theoretical discussion of dimensionality and overfitting is provided in Section 3.6, and extended validation experiments, including additional parameter stress tests, are presented in the appendices.

### 4.9 Summary of empirical findings

Taken together, the SATSA and Dental analyses show that:

- Entropy trajectories derived from longitudinal Likert-scale data are numerically stable and smooth enough to serve as inputs and evaluation targets for continuous-time models.

- A compact, coupled ODE system initialized from the first-wave entropy values can reproduce broad cohort-level trends for two distinct datasets under fixed parameterizations.

- Out-of-sample errors from leave-one-wave-out validation remain moderate, given the small number of timepoints, and support the view that the model captures structured variation rather than purely noise.

- Parameter sensitivity and multistart analyses indicate that several parameters are meaningfully constrained by the data, while others remain weakly identifiable under the present sample sizes, suggesting clear directions for future refinement.

The primary contribution of these results is methodological: they demonstrate that pooled entropy indices from longitudinal cohort data can be linked to a low-dimensional, autonomous ODE system in a way that is numerically stable, empirically grounded, and reproducible across distinct datasets.

## 5 Discussion

The present work introduced a minimal, entropy-initiated dynamical framework (ECTO) for modeling longitudinal cohort data. By compressing item-level response distributions into pooled entropy indices and using those indices to initialize a low-dimensional system of coupled ordinary differential equations, we showed that it is possible to reproduce broad cohort-level trajectories in two distinct datasets (SATSA and a U.S. dental student cohort) using a single, interpretable model class. The focus is methodological: the goal is not to propose a psychological or biological theory of any specific trait, but to demonstrate that information-theoretic preprocessing can be linked coherently to continuous-time modeling in a way that is numerically stable, reproducible, and empirically grounded.

Across both datasets, the model generates continuous trajectories that align reasonably well with observed entropy time series under a compact parameterization. The SATSA analysis illustrates that entropy-compressed psychometric data over six waves can be approximated by an autonomous ODE system initialized at the first wave, while the Dental dataset shows that the same structural form can be transferred to a different cohort, with different items and fewer timepoints, and still maintain moderate predictive performance. Leave-one-wave-out validation and parameter sensitivity analyses support the view that the model is capturing structured variation rather than trivially interpolating the data, despite the small sample of timepoints.

A practical benefit of the entropy-based approach is its robustness to attrition. As demonstrated in the supplementary analyses, entropy trajectories remain well-defined even as the number of respondents declines over time. Because each entropy value aggregates information from the full response distribution at a given wave, the signal can remain stable and interpretable under conditions where raw counts become sparse or uneven. This suggests that entropy-based compression may provide a principled way to extract longitudinal structure from legacy datasets that would otherwise be considered too incomplete for conventional modeling.

At the same time, several limitations are important to acknowledge. First, the number of timepoints in both datasets is limited (six and four waves, respectively), which constrains the strength of any conclusions about model adequacy and parameter identifiability. Second, the current implementation uses deterministic, autonomous dynamics with fixed parameters between waves; uncertainty, individual-level variability, and time-varying covariates are not yet modeled. Third, although multistart and local sensitivity analyses indicate that some parameters are meaningfully constrained by the data, others remain weakly identified in this setting, and alternative parameterizations may yield comparable fits. Finally, the analyses focus on a small number of items per dataset and do not yet explore higher-dimensional or multi-trait versions of the framework.

Overall, the results should be interpreted as an initial demonstration that entropy trajectories from longitudinal cohort studies can be treated as dynamical objects and linked to a compact ODE system, rather than as a final model of any particular psychological or biological process. The value of the framework lies in showing that such a link is technically feasible and empirically nontrivial, and that it can be implemented in a fully reproducible way.

## 6 Conclusion

This paper introduces a novel modeling framework, Entropy-initiated Coupled-Trait ODEs (ECTO), for modeling behavioral data using information-theoretic preprocessing. By using entropy-derived indices to initialize and evaluate a system of nonlinear differential equations, the approach connects information-theoretic preprocessing with continuous-time dynamical modeling in a transparent and reproducible way. In this formulation, entropy serves as a compressed, population-level descriptor through which trait dynamics can be modeled as recursive, continuous-time processes.

Even in its first-generation form, the model demonstrates stable qualitative behavior and predictive alignment with entropy-derived psychometric time series. The coupled dynamics of saturating, coupled, and feedback-driven interactions yield emergent structure consistent with constrained trait persistence. While the model does not make psychological claims per se, it shows that entropy-compressed behavioral data can be integrated into a phenomenologically structured dynamical framework, offering an interpretable alternative to black-box methods.

The present work is intentionally minimalistic, serving as a proof of concept for a more general program. The long-term aim is to establish a viable formalism for interdisciplinary research spanning psychometrics, ecology, evolutionary biology, and information theory. Future iterations will refine identifiability, integrate stochasticity, expand to multivariate trait systems, and explore machine learning interfaces for navigating high-dimensional parameter landscapes.

In sum, this work establishes a minimal, interpretable framework for integrating entropy-compressed behavioral data with autonomous dynamical systems, providing a foundation for future methodological development.

### 6.1 Code availability

Key codes used in this study are publicly available in 'walk-through' format in the *ECTO GitHub Repository* located and available for download at: https://github.com/amr28693/ECTO_walkthrough_2026

This repository includes a modeling notebook which demonstrates: entropy extraction from raw Likert-scale data; ODE-based trait simulations under the ECTO framework (including replication from the main text); an iteration of ECTO where the G term is given degrees of freedom; as well as additional validation in the form of analysis of the Dental dataset by Leite et al.

The modeling notebook provided as supplementary material uses the Jupyter Notebook framework [20] to reproduce results and allow parameter exploration.

### 6.2 Computational implementation and reproducibility

All preprocessing, entropy calculations, and numerical simulations were performed in Python. Data manipulation and aggregation were conducted using pandas [21] and NumPy [22]. Numerical integration and analysis were executed in a reproducible Jupyter notebook environment, and figures were generated using Matplotlib [23]. All code and dependencies are provided in the accompanying public repository.

## Supporting information

**S1 Appendix. Supplemental appendices.**
(PDF)

## Author contributions

**Conceptualization:** Anderson M. Rodriguez.

**Data curation:** Anderson M. Rodriguez.

**Formal analysis:** Anderson M. Rodriguez.

**Funding acquisition:** Anderson M. Rodriguez.

**Investigation:** Anderson M. Rodriguez.

**Methodology:** Anderson M. Rodriguez.

**Project administration:** Anderson M. Rodriguez.

**Resources:** Anderson M. Rodriguez.

**Software:** Anderson M. Rodriguez.

**Supervision:** Anderson M. Rodriguez.

**Validation:** Anderson M. Rodriguez.

**Visualization:** Anderson M. Rodriguez.

**Writing – original draft:** Anderson M. Rodriguez.

**Writing – review & editing:** Anderson M. Rodriguez.

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
