## [Decision Letter · Decision Letter 0]

29 Aug 2025

Dear Dr. Rodriguez,

Thank you for submitting your manuscript to PLOS ONE. After careful consideration, we feel that it has merit but does not fully meet PLOS ONE’s publication criteria as it currently stands. Therefore, we invite you to submit a revised version of the manuscript that addresses the points raised during the review process.

Your manuscript was well received by two reviewers, who praised its creative and interdisciplinary approach. However, they also identified several issues that must be addressed prior to further consideration. In particular, they recommend clarifying key definitions and methodological details, tempering causal claims, providing stronger validation and benchmarking, and addressing risks related to sample size and parameter identifiability. The presentation should also be improved, with the main text focused on core findings and supplementary materials used for secondary analyses. Reviewers further suggested improvements to the code documentation to enhance reproducibility. We encourage you to revise accordingly and resubmit once these points have been addressed.

We look forward to receiving your revised manuscript.

Kind regards,

Mario Treviño Villegas, Ph.D

Academic Editor

PLOS ONE

2. Please update your submission to use the PLOS LaTeX template. The template and more information on our requirements for LaTeX submissions can be found at http://journals.plos.org/plosone/s/latex .

Additional Editor Comments:

Reviewer #1:

Reviewer #2:

Reviewers' comments:

Reviewer's Responses to Questions

**Comments to the Author**

1. Is the manuscript technically sound, and do the data support the conclusions?

Reviewer #1: Partly

Reviewer #2: Partly

2. Has the statistical analysis been performed appropriately and rigorously?

Reviewer #1: Yes

Reviewer #2: No

3. Have the authors made all data underlying the findings in their manuscript fully available?

Reviewer #1: Yes

Reviewer #2: Yes

4. Is the manuscript presented in an intelligible fashion and written in standard English?

Reviewer #1: Yes

Reviewer #2: Yes

Reviewer #1: The authors present an interesting study on the ability of the Ordinary Differential Equation framework to model Likert-scale psychometric data over time. Their analysis is well conceived and statistically sound.

I would recommend that the authors conduct a limited study on a validation dataset that is similar to the original dataset from the Swedish Adoption/Twin Study on Aging dataset to establish whether the model is generalizable to other datasets that measure similar variables. The authors should discuss potential limitations of the types of data that could be implemented into this framework of analysis to demonstrate where their methods are applicable. This would allow for the conclusions that the authors draw to be more thoroughly founded and validated to determine whether their framework is generalizable.

Reviewer #2: Firstly, I wasn’t to applaud the (single?) author for tackling an interesting interdisciplinary topic with mathematical tools. I especially appreciate the author making available the code they used and the transparency of the results. Furthermore, the use of ODEs instead of machine learning makes this research much more useful. I thought the work is creative and spans psychometrics, dynamical systems, and evolutionary theory. I’m a physicist with biological modelling training. Hopefully a person that understands psychometrics is involved in this review. I will try to be helpful in trying to get the best possible paper out of this submission, so please take my bluntness as honest help.

Let’s start with the main concept of neuroticism. Neuroticism is broad and overlaps with anxiety, depression, stress reactivity. i.e a “catch-all” for negative affect. Can the author please include the definition of the concept they are using in this research in the introduction? From the methodology, this seems to be a score from a statistical composite trying to capture the individual’s tendency to experience negative emotions such as anxiety, sadness, irritability, or emotional instability (not direct biological measurement). Stating this from the start, will help readers decide whether they wasn’t to read the whole paper or not. From this review, I am operating under the assumption that neuroticism is not a mechanistic unit of biology, but rather a psychometric abstraction.

On to the ODE’s: Could you please add clear definitions of each term? how exactly do they encode mutation–selection balance, pleiotropy, metabolic constraints, and environmental feedback? Without the explicit form, the claims remain metaphoric. If this is the intention, please state that.

On to entropy: Entropy (bits) and time (years, irregular intervals) have different units. Could the author please provide detail on the parameter units, scaling or nondimensionalization to make the model interpretable and comparable? It’s unclear to me if the entropy is computed at each item level, across respondents, or pooled across items. This choice critically affects interpretation. Could the author please clarify whether entropy reflects individual dispersion or population-level heterogeneity? Also, if entropy is being treated as a mechanistic input (vs a descriptive output), this must be tested through surrogate or intervention experiments—e.g., replacing entropy with randomised traces and showing loss of emergent dynamics. The author should soften their claims about entropy being a “causal driver” unless supported by intervention-style analyses.

On to sample size: Given the small sample of timepoints (SATSA waves), parameter values may be poorly constrained. If possible, could the author include identifiability diagnostics (e.g., profile likelihoods, multistart fits, parameter correlation matrices)? If not they could state that there are risks regarding sensitivity due to small sample, and that future work could consider small-sample bias corrections.

On to parameters: Parameter identifiability given six SATSA waves is not strong. The author could consider diagnostics such as profile likelihoods, parameter correlation matrices, or sloppiness analyses to provide stronger parameter identification.

On to statistics: The author uses fit statistics like RMSE and R^2 over six points, which offers some, evidence for their prediction claims. To provide stronger evidence, the author could implement out-of-sample testing, e.g., leave-one-wave-out, to assess forecast performance.

On to the attractor state claims: This is the weakest part of the document, as the attractor evidence is quite thin. Usually, multi-stability claims are supported by bifurcation analyses, phase portraits, and stability basin identification—not just narrative assertion. If the author wants to keep this as evidence that supports the claims, demonstrations of how varying parameters shift attractor landscapes are needed. A similar situation occurs with Lyapunov-Proxy Validity. With only a handful of timepoints, estimating divergence rates is quite unreliable. The authors should validate their Lyapunov-proxy metric on synthetic systems with known dynamics and sampling patterns, or soften their claims. I think the ODEs have proved enough, and this could be future work. You don’t need to use complex techniques at this point, and I think it takes away from the main message of the paper.

On to the validation: Benchmarking against an uncoupled null system is too weak; coupling almost always improves fit. Would the authors consider out-of-sample forecasting (leave-one-wave-out), to provide stronger evidence of predictive skill?

Further clarifications: “Environment” is described as a recursive driver but is not defined or measured. What specific environmental variable(s) are used, with what lag structure?

To justify entropy as a functional driver rather than a descriptive marker, surrogate analyses (e.g., shuffled entropy series) should be used to test whether coupling structure persists.

Also, in Line 67: Can you please explain what you mean by a lawful, coherent system?

On to documentation: I thought the GitHub repository was much better for me to understand what the author was doing (https://github.com/amr28693/ECTO_system_walkthrough). I will point out there’s a mismatch between what's documented (README notebooks) and what's actually present in the repository, which was a bit confusing. I encourage the author to align the pipeline, labelling, and documentation across branches. Also, I couldn’t find an environment specification (e.g., a requirements.txt) nor the preprocessing pipeline to convert ICPSR data into entropy trajectories. Could the author please add these for reproducibility?

Finally, suggestions on shortening the 120 page manuscript: I think this is a very original piece of research, but the length of the manuscript and the highly abstract descriptions detract from the main finding. To turn it into a sharp, publishable article, the author should trim aggressively to highlight the core conceptual contribution and leave secondary explorations for appendices or follow-ups. For a journal such as Plos One, I would expect a 20–25 pages document.

I would suggest the following pipeline:

1 – Identify the Core Contribution

Define the one main idea: ECTO = entropy-reduced psychometric trajectories embedded in coupled ODEs that exhibit structured attractor dynamics.

Clarify novelty: not “I did everything,” but I show entropy can serve as a dynamical variable that yields attractor behaviour for a long-standing trait (neuroticism).

Choose one flagship dataset: focus on SATSA neuroticism (or one trait), not multiple traits or ecology extensions.

2 – Streamline Structure

Abstract & Introduction (2–3 pages)

One motivating puzzle (trait stability despite maladaptation).

One key contribution (ECTO framework).

One headline result (multi-stable attractors in SATSA neuroticism).

Methods (6–8 pages) depicting Data pipeline and ODE formulation (equations and biological interpretation)

Fitting procedure: parameter estimation, identifiability checks.

Baseline comparisons: uncoupled ODE + simple time-series models.

Results (6–8 pages) including: Empirical entropy trajectories, Coupled vs uncoupled fits and Evidence for attractor structure (phase portrait, bifurcation diagram, or just remove this).

Forecasting or out-of-sample validation. Either validate rigorously Lyapunov proxy analysis or cut.

Discussion (4–5 pages): Include here the interpretation of attractors in trait stability and the limits of entropy as a “driver.” Discuss future extensions (machine learning, multi-omics) in one concise paragraph.

Supplement/Appendix should include secondary traits, ecology case study, Lyapunov-proxy experiments, extensive stress tests, and theory elaborations belong here, not in the main text.

3 – Cut or Relegate expository text

Compress the lengthy expositions of evolutionary theory to 1–2 paragraphs + schematic.

Move multiple traits and species/ecology comparisons to supplement or another paper.

Replace repeated justifications with one clear figure + one paragraph.

Place technical derivations in appendix or online notebook.

Final Recommendation: Please condense the manuscript into a single strong narrative around entropy trajectories + coupled ODE attractor dynamics in SATSA neuroticism. Push all auxiliary analyses (ecology, other traits, proxy Lyapunov) to supplementary material or spin-off manuscripts. I would strongly encourage the author to first submit some of this material, perhaps as a Late Breaking Abstract, at a conference such as ALIFE in the Models of Consciousness track or ABMHuB’25. This work would be very welcome there and the author might find a suitable collaborator to balance out some of the interdisciplinary work and claims in this work.

**Do you want your identity to be public for this peer review?** For information about this choice, including consent withdrawal, please see our Privacy Policy

Reviewer #1: No

Reviewer #2: No

---

## [Author Response · Author response to Decision Letter 1]

30 Jan 2026

Response to Reviewers

I sincerely thank the Academic Editor and both reviewers for their careful evaluation of the manuscript and for their constructive and detailed feedback. I have revised the manuscript ex- tensively to address all concerns, and in turn have hopefully clarified the scope and interpretation of the proposed framework, and strengthened validation and robustness analyses. Below, I respond point-by-point to each comment raised by the Academic Editor and reviewers, with references to specific sections of the revised manuscript and appendix.

Reviewers’ Responses to Editorial Questions

1. Is the manuscript technically sound, and do the data support the conclusions?

Reviewer #1: Partly Reviewer #2: Partly

The manuscript has been substantially revised to strengthen the technical grounding and empir- ical support for the conclusions. Key revisions include the addition of an independent validation dataset (a U.S. dental student longitudinal cohort) to assess generalizability beyond the SATSA dataset (Sections 4.5–4.6), the implementation of leave-one-wave-out (LOO) out-of-sample valida- tion for both datasets (Sections 4.3 and 4.6), and expanded robustness analyses including multi- start optimization, parameter sensitivity sweeps, null model comparisons, and attrition stress tests (Sections 4.7–4.8; Appendix B–D). Several exploratory analyses (marked-up in green font in track- changes-manuscript) have been moved to the appendix or explicitly framed as illustrative rather than definitive.

2. Has the statistical analysis been performed appropriately and rigorously? Reviewer #1: Yes Reviewer #2: No

To address concerns regarding statistical rigor, I have added explicit out-of-sample forecasting via leave-one-wave-out validation for both datasets (Sections 4.3 and 4.6). The coupled ECTO system is now benchmarked against baseline and null models, including uncoupled dynamics and flat entropy controls (Section 4.4; Appendix C). Parameter identifiability and sensitivity are examined through multistart fitting and parameter sweep analyses (Section 4.7; Appendix C–D), and the limitations imposed by sparse longitudinal sampling are discussed explicitly in the Discussion (Section 5).

3. Have the authors made all data underlying the findings fully available? Reviewer #1: Yes Reviewer #2: Yes

All data underlying the findings are made available in accordance with PLOS ONE data policy. Publicly available datasets are cited and linked, and all preprocessing, analysis, and simulation code required to reproduce the results is provided in the accompanying repository. Any restrictions associated with third-party data sources (e.g., ICPSR-hosted data) are described explicitly in the Data Availability Statement.

4. Is the manuscript presented in an intelligible fashion and written in standard English? Reviewer #1: Yes Reviewer #2: Yes

The manuscript has been edited for clarity, concision, and consistency. The main text has been sub- stantially condensed, with secondary analyses and methodological details moved to the appendix.

1

Reviewer Comments to Author Reviewer #1

Comment (Paraphrased): The reviewer recommends conducting a limited validation study on a dataset similar to the SATSA cohort to assess generalizability and suggests discussing potential limitations on the types of data applicable to the proposed framework.

Response: I have addressed this comment in two ways. First, I have added an independent validation analysis using a longitudinal U.S. dental student cohort, which differs from SATSA in population, measurement context, and survey structure. This dataset is analyzed using the same entropy preprocessing and ECTO modeling pipeline, with separate parameter fitting and leave-one- wave-out validation (Sections 4.5–4.6). The inclusion of this cohort provides an external test of the framework beyond the original aging twin dataset and demonstrates that the proposed approach is not specific to SATSA.

Second, I have expanded the discussion of scope and limitations to clarify the types of longitu- dinal psychometric data for which the framework is appropriate. In particular, I emphasize that the method is intended for sparsely sampled cohort-level data with repeated categorical measure- ments, and that it is not designed to recover individual-level dynamics or mechanistic biological processes (Section 5). These clarifications are intended to delimit the domain of applicability and avoid overgeneralization.

Reviewer #2

Comment (Paraphrased): The reviewer raises concerns regarding the definition and inter- pretation of neuroticism, the interpretation of the ODE terms, the role and meaning of entropy, parameter identifiability under sparse sampling, statistical validation and forecasting, claims re- garding attractors and Lyapunov-like metrics, baseline comparisons, the definition of environment, repository documentation, and manuscript length.

Response: I address these points as follows.

Definition of neuroticism: The Introduction now explicitly defines neuroticism as a psychome-

tric construct derived from questionnaire-based composite measures, rather than as a mechanistic biological entity. This clarification is stated at the outset to avoid causal or biological misinterpre- tation (Section 1).

Interpretation of the ODE terms: All state variables and parameters are now defined explicitly. Terms such as selection pressure, pleiotropy, metabolic constraint, and environmental feedback are described as phenomenological analogues rather than mechanistic claims. The manuscript states explicitly that these labels are interpretive aids and not direct biological mappings (Section 3).

Role and interpretation of entropy: Entropy is now consistently framed as a population-level summary statistic computed from Likert response distribution measured in waves, and used solely to initialize the dynamical system. Entropy is not treated as a time-varying causal driver. Claims implying causal interpretation have been softened or removed (Sections 2.1–2.4; Section 5; Ap- pendix A).

Units and scaling: Parameter units and scaling are described explicitly, and the system is presented in dimensionless form following normalization. The implications of irregular temporal sampling and nondimensionalization are discussed in the Methods (Section 3).

2

Parameter identifiability and sensitivity: Given the limited number of timepoints, I have added explicit sensitivity analyses. These include multistart optimization, two-dimensional parameter sweeps, and univariate capacity sweeps, which demonstrate that model behavior is not confined to narrowly tuned parameter values (Section 4.7; Appendix C & D). Limitations due to sparse sampling are discussed explicitly (Section 5).

Statistical validation and forecasting: Leave-one-wave-out validation has been implemented for both the SATSA and dental student datasets to assess out-of-sample predictive performance (Sec- tions 4.3 and 4.6).

Attractor and Lyapunov-related claims: Claims regarding attractor structure and Lyapunov-like behavior have been substantially softened. Exploratory analyses related to these concepts have been moved to the appendix and are framed as illustrative rather than definitive. No strong claims of multistability or rigorous dynamical invariants are made in the main text (Section 5; Appendix C).

Baseline comparisons: The coupled ECTO system is benchmarked against simpler alternatives, including uncoupled dynamics and flat entropy baselines. These comparisons are reported explic- itly using RMSE and R2, with limitations of variance-based metrics discussed where appropriate (Section 4.4; Appendix C.1).

Definition of environment: The environmental term is now defined explicitly as an aggregated constraint state within the phenomenological model, rather than as a measured external variable. Its role and limitations are clarified to avoid misinterpretation as an empirically observed environ- mental signal (Section 3).

Reproducibility and repository structure: The repository has been aligned with the documented analysis pipeline. Preprocessing steps, analysis scripts, and dependencies are clearly labeled, and environment specifications have been added to support reproducibility. The repository now has a ’A’ and ’B’ tracks, each of which explore a different dataset, respectively.

Manuscript length and scope: The manuscript has been substantially condensed. Secondary analyses, exploratory extensions, and theoretical elaborations have been moved to the supplemental appendices, and the main text now focuses on the core methodological contribution and primary validation results.

---

## [Decision Letter · Decision Letter 1]

16 Feb 2026

An Entropy-Initiated Coupled-Trait ODE Framework for Modeling Longitudinal Cohort Dynamics

PONE-D-25-39290R1

Dear Dr. Rodriguez,

We’re pleased to inform you that your manuscript has been judged scientifically suitable for publication and will be formally accepted for publication once it meets all outstanding technical requirements.

Kind regards,

Mario Treviño Villegas, Ph.D

Academic Editor

PLOS One

Additional Editor Comments (optional):

Reviewers' comments:

Reviewer's Responses to Questions

**Comments to the Author**

Reviewer #1: All comments have been addressed

Reviewer #2: All comments have been addressed

2. Is the manuscript technically sound, and do the data support the conclusions?

Reviewer #1: Yes

Reviewer #2: Yes

3. Has the statistical analysis been performed appropriately and rigorously?

Reviewer #1: Yes

Reviewer #2: Yes

4. Have the authors made all data underlying the findings in their manuscript fully available?

Reviewer #1: Yes

Reviewer #2: Yes

5. Is the manuscript presented in an intelligible fashion and written in standard English?

Reviewer #1: Yes

Reviewer #2: Yes

Reviewer #1: The authors have significantly strengthened the conclusions, reproducibility, and generalizability of the methodology and findings of the study by their use of an external dataset for validation. The ability of the methods to be applied to other types of data is now qualified in a way that enhances the reader's understanding of the applicability of the model to specific data types.

Reviewer #2: Adding an independent longitudinal cohort (U.S. dental students) had significantly elevated this research. I previously asked the author to focus on their core finding: entropy can serve as a dynamical variable that yields attractor behaviour for a long-standing trait (neuroticism). To tackle this, the author has clearly defined neuroticism as a psychometric construct, not a biological mechanism. Further changes such as ODE terms framed as phenomenological analogues, not mechanistic evolutionary claims, and entropy softened from “driver” to “summary statistic used for initialisation have helped clearly define the author’s contribution.

The author included a separate fitting and LOO validation, demonstrating the method is not SATSA-specific. Leave-one-wave-out forecasting on both datasets is a great validation. I am happy to see some predictive power of this model. I also thank the author for softening the strong claims about multistability and Lyapunov exponents, and especially for moving these analyses to appendices as illustrative. While still a very large document, this has focused the narrative and tightened the results.

I will warn that, even with LOO validation and sensitivity sweeps, some will remain sceptical that 5–6 timepoints can meaningfully constrain ODE dynamics. However, because the researcher has explicitly acknowledged this limitation, this shouldn’t stop this from being published as the claims are now proportional to evidence. I am very impressed the author was actually able to reduce the size of the manuscript to a reasonable length, while clarifying many ideas that were very nebulous in the previous version, while strengthening the original contribution. Once again, I would like to applaud the researcher for a very interesting piece of work and wish them all the best on their future endeavours.

**Do you want your identity to be public for this peer review?** For information about this choice, including consent withdrawal, please see our Privacy Policy

Reviewer #1: No

Reviewer #2: No

---

## [Editor Report · Acceptance letter]

PONE-D-25-39290R1

PLOS One

Dear Dr. Rodriguez,

I'm pleased to inform you that your manuscript has been deemed suitable for publication in PLOS One. Congratulations! Your manuscript is now being handed over to our production team.

Kind regards,

on behalf of

Dr. Mario Treviño Villegas

Academic Editor

PLOS One